# MindAligner: Explicit Brain Functional Alignment for Cross-Subject Visual Decoding from Limited fMRI Data

Yuqin Dai[*2,1,4]   Zhouheng Yao[*1]   Chunfeng Song[1]   Qihao Zheng[1]   Weijian Mai[1]
Kunyu Peng[5]   Shuai Lu[4]   Wanli Ouyang[1,3]   Jian Yang[†2]   Jiamin Wu[†1,3]

https://github.com/Da1yuqin/MindAligner

## Abstract

Brain decoding aims to reconstruct visual perception of human subject from fMRI signals, which is crucial for understanding brain's perception mechanisms. Existing methods are confined to the single-subject paradigm due to substantial brain variability, which leads to weak generalization across individuals and incurs high training costs, exacerbated by limited availability of fMRI data. To address these challenges, we propose MindAligner, an explicit functional alignment framework for cross-subject brain decoding from limited fMRI data. The proposed MindAligner enjoys several merits. First, we learn a Brain Transfer Matrix (BTM) that projects the brain signals of an arbitrary new subject to one of the known subjects, enabling seamless use of pretrained decoding models. Second, to facilitate reliable BTM learning, a Brain Functional Alignment module is proposed to perform soft cross-subject brain alignment under different visual stimuli with a multi-level brain alignment loss, uncovering fine-grained functional correspondences with high interpretability. Experiments indicate that MindAligner not only outperforms existing methods in visual decoding under data-limited conditions, but also provides valuable neuroscience insights in cross-subject functional analysis.

## 1. Introduction

The brain serves as the center of human cognition and unraveling its underlying mechanisms holds profound academic significance (Naselaris et al., 2011). To investigate the brain's perceptual mechanisms, functional magnetic resonance imaging (fMRI) (Naselaris et al., 2011) has been widely used due to its noninvasive acquisition and precise localization of the functional regions. The advances in fMRI facilitate the research on brain visual decoding, which aims to recover visual stimuli seen by humans from their brain activity, contributing to the progress of cognitive science research and Brain-Computer Interfaces (BCI) (Qian et al., 2020; Horikawa & Kamitani, 2017).

Despite the success in fMRI-based visual decoding, the majority methods (Seeliger et al., 2018; Lu et al., 2023; Ozcelik & VanRullen, 2023) are confined to the less practical single-subject paradigm, where a customized decoding model is trained for each person subject. Due to substantial brain differences among subjects, the decoding model trained on one subject cannot be effectively transferred to others, limiting its practicality in BCI and clinical applications. In fact, variations in individual cognitive patterns and brain structures, result in significant fMRI differences (Naselaris et al., 2011). Moreover, the high acquisition cost of fMRI limits the data availability for new subjects. Thus, adapting brain decoding models to new subjects with limited data is crucial.

To address the cross-subject issue, several methods (Scotti et al., 2024b; Li et al., 2025; Wang et al., 2024) adopt brain alignment techniques in an implicit manner. They align fMRI signals from different subjects to a latent space that is assumed to capture common cognition patterns across subjects by learning subject-specific parameters. However, this implicit alignment approach has two limitations. (1) **Insufficient brain alignment**: learning a shared space that effectively aligns all subjects remains challenging due to noisy and limited fMRI data. As individuals have vast brain differences even when viewing identical stimuli, enforcing all subjects to be simultaneously aligned to a single latent space is prone to suboptimal alignment and compromised representation. Even with extensive multi-subject fMRI training, the shared latent space shows limited generalizability to unseen subjects. For instance, Unibrain (Lei et al., 2023) shows  50% performance drop when transferring the

---

[*]Equal contribution.   [1]Shanghai Artificial Intelligence Laboratory [2]Nanjing University of Science and Technology [3]The Chinese University of Hong Kong [4]Tsinghua University [5]Karlsruher Institut für Technologie.   Correspondence to:  Jian Yang <csjyang@njust.edu.cn>, Jiamin Wu <jiaminwu@cuhk.edu.hk>.

*Proceedings of the 42nd International Conference on Machine Learning*, Vancouver, Canada. PMLR 267, 2025. Copyright 2025 by the author(s).

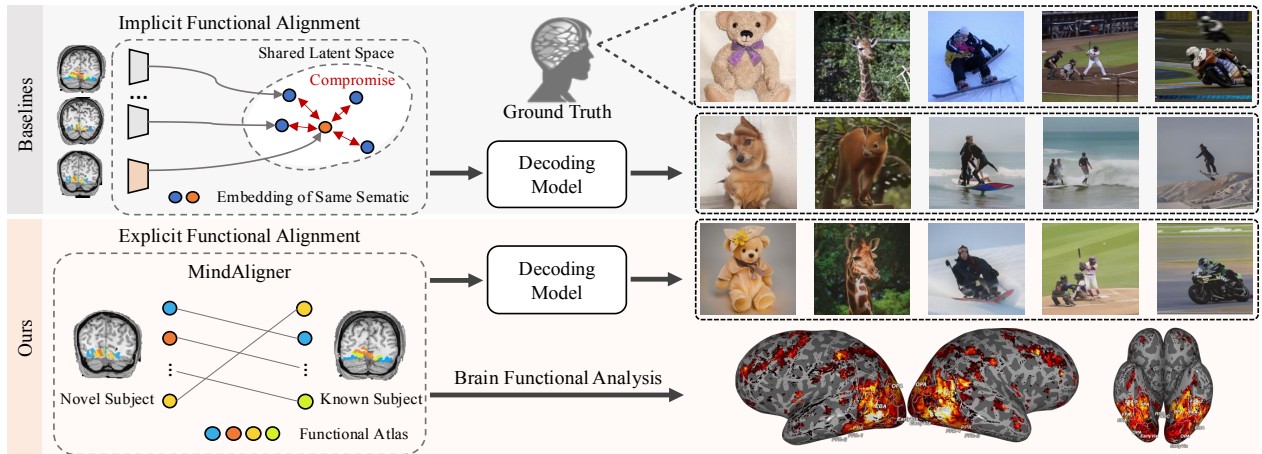

*Figure 1.* Different approaches to functional alignment in brain decoding: Prior works (Scotti et al., 2024b; Li et al., 2025) adopt implicit alignment approach that aligns all subjects into a single latent space, which may lead to suboptimal alignment. Differently, MindAligner employs an **explicit alignment strategy**, mapping novel subject signals to seen ones by establishing fine-grained functional correspondences. MindAligner not only enables high-quality visual reconstruction from fMRI signals but also facilitates brain functional analysis across subjects.

decoding model to new subjects. (2) **Lack of functional interpretability**: existing latent alignment methods (Scotti et al., 2024b; Wang et al., 2024; Li et al., 2025) fail to explicitly account for cognitive pattern relationships between subjects. Their alignment is incapable of localizing the brain regions for functional differences and commonalities. This lack of interpretability not only limits cross-subject knowledge transfer in new subject adaptation, but also hinders analysis of neural functional mechanisms underlying human perception process. Given these limitations, an important question arises: *can we create a brain alignment framework enabling effective new-subject adaptation and brain functional analysis under data scarcity?*

To answer this question, our motivation is to establish an explicit brain functional alignment framework that maps the novel subject's signal to a seen subject's signal. Given an arbitrary new subject, our explicit alignment can establish fine-grained functional correspondences between the new subject and seen subjects in the original brain space, as shown in Fig. 1. The aligned fMRI signal can not only seamlessly be integrated into the pre-trained decoding model of seen subjects but also reveals brain region-level cross-subject variability. However, achieving such brain alignment is challenging, as it requires paired fMRI from subjects performing the same task (*i.e.*, viewing identical visual stimuli (Bazeille et al., 2021)), a condition not met by the existing dataset (Allen et al., 2022).

Based on the above observations, we propose **MindAligner**, an explicit functional alignment framework for cross-subject visual decoding with limited fMRI data. The core of our method is to train a cross-subject **Brain Transfer Matrix** (BTM) that projects the brain signals of a new subject to

one of the known subjects in the voxel-level. To overcome the lack of strictly paired fMRI signals, we propose a **Brain Functional Alignment module** (BFA) to perform soft cross-subject alignment between fMRI signals from different but similar visual stimuli, facilitating the mapping of functionally equivalent cortical areas. Specifically, BFA first decomposes the brain transfer matrix into two low-rank linear layers, enhancing parameter efficiency to facilitate effective adaptation with limited data. In the latent space of the BTM, a cross-stimulus neural mapper is designed to transform the fMRI under different visual stimuli, with stimulus differences as mapping condition. To achieve sufficient and fine-grained alignment, we design a multi-level brain alignment losses that incorporates a signal-level reconstruction loss and a latent alignment loss guided by visual semantic similarities. In this way, the resulting brain transfer matrix not only facilitates fine-grained alignment without shared stimuli constraint, but also encodes cross-subject brain relations for enhanced functional interpretability and neuroscience analysis.

In summary, our contributions are as follows:

- We propose MindAligner, the first explicit brain alignment framework that enables cross-subject visual decoding and brain functional analysis in the data-limited setting.

- We propose a brain transfer matrix to establish fine-grained functional correspondences between arbitrary subjects. This matrix is optimized through a brain functional alignment module, which employs a multi-level alignment loss to enable soft cross-subject mapping.

- Experiments demonstrate that MindAligner outper-

forms state-of-the-art methods in visual decoding with only 6% of the whole model's parameters learned.

• We conduct cross-subject brain functional visualization and discover that the early visual cortex shows similar activities across subjects, while the higher visual cortex related to memory and spatial navigation exhibits significant inter-subject variability.

## 2. Related Work

### 2.1. fMRI-Based Brain Decoding

Brain decoding seeks to reconstruct the visual stimuli perceived by subjects based on their brain activity, offering a deeper understanding of the brain's mechanisms for processing external information (Naselaris et al., 2011). Earlier work (Horikawa & Kamitani, 2017) reveals a correlation between Deep Neural Networks (DNNs) image representations and neural activity in the visual cortex using sparse linear regression. With the advent of generative models (Goodfellow et al., 2020; Ho et al., 2020) and extensive fMRI datasets (Allen et al., 2022), visual decoding has shifted towards mapping brain signals to the latent spaces of large models, facilitating the reconstruction of diverse visual stimuli (Gu et al., 2022; Ozcelik et al., 2022; Gao et al., 2024a; Zhou et al., 2024; Mai et al., 2024; Gao et al., 2024b). This approach has proven effective in utilizing latent diffusion models for image reconstruction (Lin et al., 2022; Takagi & Nishimoto, 2023; Mai & Zhang, 2023; Scotti et al., 2024a; Chen et al., 2023), addressing inter-subject differences by either training separate models for individual subjects or employing partially unified models with subject-specific parameters. However, influenced by neural variability, cross-subject brain signals in the latent space are prone to semantic conflicts, which can lead to convergence at suboptimal points. MindAligner addresses this by leveraging an explicit functional alignment framework across brains, this approach more effectively utilizes shared functionalities among subjects, thereby mitigating semantic conflicts.

### 2.2. Cross-Subject Functional Alignment

As brains differ both in size and processing mechanisms (Allen et al., 2022; Finn et al., 2017), the resulting variability in fMRI signals has spurred research into brain alignment methods. The ideal condition for functional alignment methods often depends on **shared stimuli**, requiring paired data from multiple subjects exposed to identical visual inputs, with alignment achieved through reconstruction loss optimization (Dadi et al., 2020; Rastegarnia et al., 2023). Relying on this ideal shared stimuli condition, a new paradigm (Bazeille et al., 2019; Thual et al., 2022; 2023; Ferrante et al., 2024) has emerged, enabling explicit alignment under shared stimuli by transforming one subject's

signal to that of another, thereby preserving functionality and facilitating knowledge transfer across subjects. However, such ideal large-scale shared stimuli are often absent. To better adapt to these scenarios, current methods focus on either anatomical alignment (Bao et al., 2025; Jiang et al., 2024; Shen et al., 2024) or functional alignment in latent space (Scotti et al., 2024a;b; Wang et al., 2024; Gong et al., 2025). MindEye2 (Scotti et al., 2024b) employs ridge regression to align different subjects into a shared latent space, followed by a shared decoding module. MindBridge (Wang et al., 2024) generates pseudo stimuli to create shared stimuli for brain alignment. However, these alignment methods either rely on shared stimuli, restricting their applicability to datasets without such conditions, or utilize latent space alignment, which impedes their ability to uncover inter-subject neural variability. We introduce an explicit brain functional alignment model to conquer the restriction of the shared stimuli and enhance the interpretation of inter-subject neural variability.

## 3. Preliminary

We begin with the illustration of the problem definition, and preliminary on cross-subject brain decoding baseline that reconstructs visual stimuli in the data-limited setting.

**Problem Definition.** The acquisition of fMRI data is both time-intensive and costly, leading to brain decoding scenarios frequently constrained by limited data. Therefore, this study focuses on investigating cross-subject brain decoding in a data-limited setting. We follow MindEye2 (Scotti et al., 2024b) to build this setting on the Natural Scenes Dataset (NSD) (Allen et al., 2022). Specifically, the decoding model is first pre-trained for one or several subjects $S_K$ using their full 40 scanning sessions of fMRI signals. Subsequently, the pre-trained decoding model is transferred to a new subject $S_N$, using only a single session of scanned fMRI (approximately 1 hour of data, representing just 2.5% of the full dataset). Finally, the adapted model is tested on 1000 images shared across all subjects for subject $S_N$.

**Cross-subject Brain Decoding Baseline.** Here we introduce our cross-subject decoding baseline method (Scotti et al., 2024b). To reduce inter-subject differences, the baseline model first employs linear layers to map brain signals from different subjects into a shared latent space. Then the fMRI embeddings are aligned with the latent space of a CLIP model (Radford et al., 2021) through a diffusion prior (Ramesh et al., 2022), thereby leveraging generative models' capabilities for visual reconstruction. The output embeddings are then fed through a low-level submodule and a retrieval submodule. Two corresponding losses are utilized: a low-level reconstruction loss $\mathcal{L}_{low-level}$ between the blurry images generated by the low-level submodule and the ground truth, and a bidirectional MixCo loss $\mathcal{L}_{BiMixCo}$

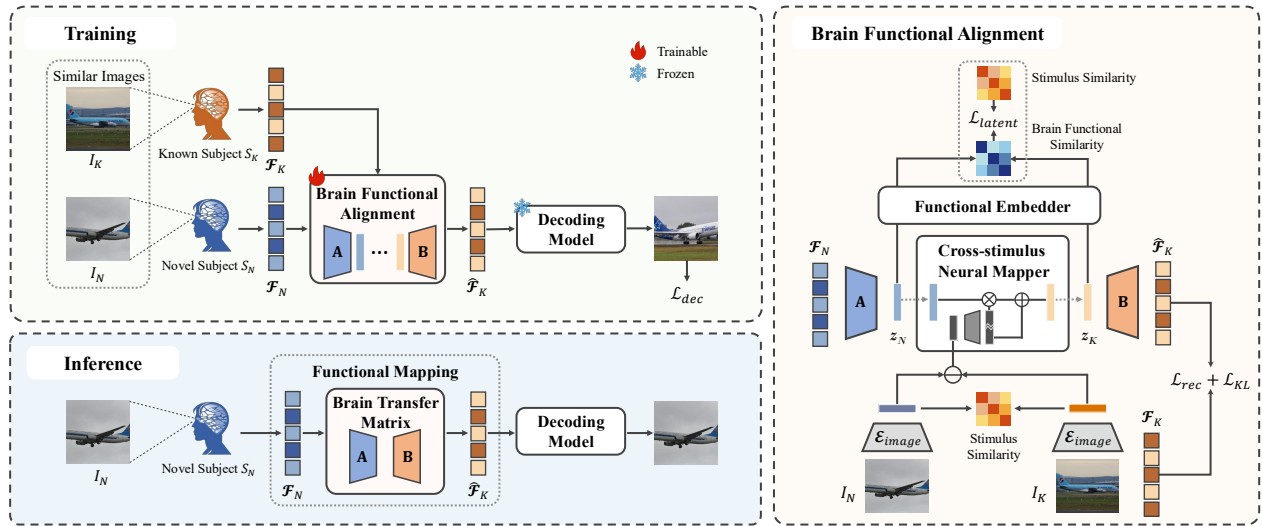

*Figure 2.* Overview of MindAligner. To achieve explicit brain functional alignment, given a pre-trained brain decoding model, we design a **Brain Functional Alignment Module (BFA)** that learns a Brain Transfer Matrix (BTM) $\mathcal{M}$ for fMRI mapping between the known and novel subjects. BTM is decomposed into two low-rank matrices $\mathbf{A}$ and $\mathbf{B}$ to create latent space for further alignment. The Cross-Stimulus Neural Mapper is proposed to create fMRI pairs under shared stimuli. In addition to the alignment losses $\mathcal{L}_{rec}$ and $\mathcal{L}_{KL}$ between generated and real fMRI, a latent alignment loss $\mathcal{L}_{latent}$ guides functional alignment based on stimulus similarities. In the inference stage, only the BTM is utilized for functional mapping, enabling cross-subject brain decoding.

to perform contrastive optimization between the retrieval module's output and the CLIP image embeddings. The final loss for the decoding model's training is formulated as: $\mathcal{L}_{dec} = \mathcal{L}_{prior} + \alpha_1 \mathcal{L}_{low-level} + \alpha_2 \mathcal{L}_{BiMixCo}$, where $\mathcal{L}_{prior}$ denotes the diffusion prior loss that measures discrepancies between the CLIP image embedding and the outputs produced by the diffusion prior. More details can be found in (Scotti et al., 2024b).

## 4. MindAligner

### 4.1. Overview

Building on the pre-trained brain decoding model, MindAligner utilizes a **Brain Transfer Matrix** (BTM) to transform signals from novel subjects into the signal space of a known subject with limited data, thereby enabling cross-subject brain decoding. To achieve reliable and fine-grained brain alignment, we design the **Brain Functional Alignment Module** (BFA). Notably, the alignment module is utilized only during the training phase to assist BTM learning; during the inference phase, only the lightweight BTM is retained. Next, we provide a detailed illustration of each module of MindAligner.

### 4.2. Brain Transfer Matrix

Given the fMRI signal $\mathcal{F}_N$ of an arbitrary novel subject $S_N$ as input, the Brain Transfer Matrix (BTM) $\mathcal{M}$ aims to transform it into the fMRI signal of a subject $S_K$ seen

during pre-training through a linear transformation:

$$\hat{\mathcal{F}}_K = \mathcal{M} \times \mathcal{F}_N, \qquad (1)$$

where $\hat{\mathcal{F}}_K$ denotes the fMRI signal projected to the brain space of the seen subject $S_K$. To improve parameter efficiency in the data-limited setting, we decompose $\mathcal{M}$ into two low-rank matrices $\mathbf{A}$ and $\mathbf{B}$,

$$\mathcal{M} = \mathbf{A} \times \mathbf{B}, \qquad (2)$$

where $\mathbf{A} \in \mathbb{R}^{n \times h}$ and $\mathbf{B} \in \mathbb{R}^{h \times k}$, $n$ and $k$ are the voxel dimensions of the novel and known subjects' fMRI, and $h$ is the hidden dimension. The matrix decomposition creates a shared latent space between two subjects for subsequent alignment. $\mathcal{M}$ encodes transfer weights that capture region-level inter-subject brain correlations and can be utilized for functional alignment during inference. Moreover, these correlations provide valuable insights for analyzing inter-subject variability.

### 4.3. Brain Functional Alignment Module

To learn a reliable brain transfer matrix, the Brain Functional Alignment Module (BFA) conducts soft cross-subject alignment in both the brain space and the shared latent space of the BTM. As no strictly-paired fMRI data under identical stimuli is provided, we employ a cross-stimulus neural mapper to facilitate stimulus transformation, rendering fMRI-pairs under visually similar stimuli. Based on these fMRI-pairs, a multi-level brain alignment loss is employed to achieve final alignment.

*Table 1.* Visual decoding performance comparison. 1h means 1 hour of data. **Bold** indicates the best performance.

| Method | Low-Level | | | | High-Level | | | | Retrieval | |
|---|---|---|---|---|---|---|---|---|---|---|
| | PixCorr↑ | SSIM↑ | Alex(2)↑ | Alex(5)↑ | Incep↑ | CLIP↑ | Eff↓ | SwAV↓ | Image↑ | Brain↑ |
| (1 h) MindBridge | 0.109 | 0.324 | 79.0% | 86.1% | 78.7% | **82.7%** | 0.866 | 0.528 | - | - |
| (1 h) MindEye2 | 0.195 | **0.419** | 84.2% | 90.6% | 81.2% | 79.2% | 0.810 | 0.468 | 79.0% | 57.4% |
| (1 h) Ours | **0.206** | 0.414 | **85.6%** | **91.6%** | **83.0%** | 81.2% | **0.802** | **0.463** | 79.0% | **75.3%** |
| (subj1) MindBridge | 0.124 | 0.333 | 80.55% | 86.51% | 78.72% | **83.56%** | 0.868 | 0.526 | - | - |
| (subj1) MindEye2 | **0.235** | **0.428** | 88.02% | **93.33%** | **83.56%** | 80.75% | **0.798** | 0.459 | **93.96%** | 77.63% |
| (subj1) Ours | 0.226 | 0.415 | **88.19%** | 93.26% | 83.48% | 81.76% | 0.800 | **0.459** | 90.90% | **86.88%** |
| (subj2) MindBridge | 0.107 | 0.330 | 78.21% | 85.85% | 77.96% | 81.04% | 0.874 | 0.532 | - | - |
| (subj2) MindEye2 | 0.200 | **0.433** | 85.00% | 92.13% | 81.86% | 79.39% | 0.807 | 0.467 | **90.53%** | 67.18% |
| (subj2) Ours | **0.218** | 0.426 | **88.08%** | **93.33%** | **84.13%** | **82.47%** | **0.791** | **0.452** | 90.04% | **85.61%** |
| (subj5) MindBridge | 0.105 | 0.326 | 80.18% | 87.70% | 81.95% | **85.38%** | 0.840 | 0.507 | - | - |
| (subj5) MindEye2 | 0.175 | 0.405 | 83.11% | 91.00% | 84.33% | 82.53% | **0.781** | **0.444** | 66.94% | 46.96% |
| (subj5) Ours | **0.197** | **0.409** | **84.69%** | **91.61%** | **84.63%** | 82.76% | 0.784 | 0.454 | **70.62%** | **65.95%** |
| (subj7) MindBridge | 0.101 | 0.308 | 76.99% | 84.51% | 76.31% | **80.98%** | 0.882 | 0.546 | - | - |
| (subj7) MindEye2 | 0.170 | **0.408** | 80.70% | 85.90% | 74.90% | 74.29% | 0.854 | 0.504 | **64.44%** | 37.77% |
| (subj7) Ours | **0.183** | 0.407 | **81.45%** | **88.31%** | **79.92%** | 77.82% | **0.834** | **0.487** | 64.18% | **62.58%** |

**Cross-stimulus Neural Mapper.** Due to the lack of stimuli-pair where two subjects view the same image, we turn to select cross-subject fMRI pairs with similar stimuli $I_N$ and $I_K$ viewed by two subjects. The cross-stimulus neural mapper aims to transform the fMRI embedding $z_N$ under stimuli $I_N$ into those corresponding to stimuli $I_K$ of the known subject $S_K$. However, due to the absence of brain prior knowledge, directly generating fMRI signals is still challenging. Therefore, we leverage the differences between $I_N$ and $I_K$ as conditions for generating fMRI. Based on the fMRI embedding $z_N$ projected by the low-rank matrix $\mathbf{A}$, i.e., $z_N = \mathbf{A} \times \mathcal{F}_N$, we use the visual stimuli difference $E_{\text{diff}}$ viewed by the two subjects as a condition to perform linear modulation to generate the stimuli embedding $z_K$ corresponding to the stimuli $I_K$ of the known subject:

$$E_{\text{diff}} = \mathcal{E}_{image}(I_N) - \mathcal{E}_{image}(I_K), \quad (3)$$

$$z_{\text{diff}} = E_{\text{diff}} \times \mathcal{M}_{\text{diff}}, \quad (4)$$

$$z_K = \mathcal{M}_C(z_N, z_{\text{diff}}), \quad (5)$$

where $\mathcal{E}_{image}$ is the image encoder of pretrained CLIP (Radford et al., 2021). The cross-stimulus neural mapper $\mathcal{M}_C(\cdot)$ is a linear modulation that splits the condition $z_{\text{diff}}$ to scale and shift parameters using $\mathcal{M}_{\text{diff}} \in \mathbb{R}^{a \times 2h}$. $a$ is the clip embedding's dimension. These parameters can be used to modulate the input $z_N$, thereby facilitating the cross-subject stimulus transformation in the latent space. The transformed embedding $z_K$ is then projected to the known subject's space by the low-rank matrix $\mathbf{B}$, rendering a synthesized fMRI embedding $\hat{\mathcal{F}}_K$ to be aligned with the known subject's real fMRI embedding $\mathcal{F}_K$:

$$\hat{\mathcal{F}}_K = z_K \times \mathbf{B}. \quad (6)$$

The effectiveness of linear structures for cross-subject alignment is justified by both theoretical and empirical evidence (Appendix F).

**Multi-level Brain Alignment Loss.** The brain alignment loss integrates both signal-level reconstruction loss between the generated and real fMRI signals and an embedding-level alignment loss to achieve more refined alignment across different visual stimuli. To ensure the quality of synthesized fMRI $\hat{\mathcal{F}}_K$, an fMRI reconstruction loss is designed to enforce the consistency between $\hat{\mathcal{F}}_K$ and the real fMRI $\mathcal{F}_K$ of the known subject:

$$\mathcal{L}_{rec} = ||\hat{\mathcal{F}}_K - \mathcal{F}_K||_2^2. \quad (7)$$

To further enhance the alignment performance, we use the Kullback-Leibler (KL) Divergence loss to enforce the consistency between distributions of the generated and real fMRI signals:

$$\mathcal{L}_{KL} = \mathcal{KL}(\hat{\mathcal{F}}_K, \mathcal{F}_K). \quad (8)$$

To enable fine-grained functional mapping under different stimuli, we leverage intrinsic correlations between visual semantics to guide the alignment of the corresponding brain activities in the latent space. Specifically, we design a latent alignment loss $\mathcal{L}_{latent}$ by enforcing the consistency between fMRI embedding pairs and stimuli pairs:

$$\mathcal{L}_{latent} = ||(\mathcal{R}(\mathcal{E}_f(z_N), \mathcal{E}_f(z_K)) - \mathcal{R}(E_N, E_K)||_2^2, \quad (9)$$

where $E_N$ and $E_K$ denote the CLIP embeddings of $I_N$ and $I_K$. $\mathcal{R}(\cdot)$ calculates the dissimilarity matrix between embedding pairs. $\mathcal{E}_f$ denotes a functional embedder for fMRI embeddings for better dissimilarity calculation. Hence, the final brain alignment loss $\mathcal{L}_{align}$ is formulated as:

$$\mathcal{L}_{align} = \mathcal{L}_{dec} + \alpha_{rec}\mathcal{L}_{rec} + \alpha_{KL}\mathcal{L}_{KL} + \alpha_{la}\mathcal{L}_{latent}, \quad (10)$$

where $\alpha_{rec}, \alpha_{kl}, \alpha_{la}$ are the loss coefficients, and $\mathcal{L}_{dec}$ denotes the decoding loss in the baseline method. The combination of these losses can improve the semantic accuracy of visual reconstruction and also reduce the reliance on same-stimulus data.

## 4.4. Inference

During inference, only the trained BTM is used for functional alignment (Eq. 1). The generated $\hat{\mathcal{F}}_K$ is then directly fed into the pre-trained model for brain decoding. MindAligner is a lightweight functional alignment module that enables efficient cross-subject visual decoding.

## 5. Experiments

In this section, we present the implementation details, followed by fMRI-to-image reconstruction results and brain functional alignment analysis. The Appendix includes additional metrics, qualitative and quantitative results, model efficiency, and further visualizations.

### 5.1. Implementation Details

The BTM is composed of two bias-free linear layers with a hidden size $h = 4096$. The input dimension $n$ and output dimension $k$ of BTM are determined by the specific subject transfer pairs. For subjects 1, 2, 5, and 7, the dimensions are 15,724, 14,278, 13,039, and 12,682, respectively. The cross-stimulus neural mapper is implemented using the Feature-wise Linear Modulation model (Perez et al., 2018), where the input dimension of $\mathcal{M}_{\text{diff}}$ is $a = 768$, matching the CLIP embedding dimension, and the output is $2h = 8192$. The functional embedder is a linear layer with input and output sizes of $h = 4096$. The loss coefficients are set to $\alpha_{rec} = 1$, $\alpha_{la} = \alpha_{KL} = 0.001$, $\alpha_1 = 0.033$, and $\alpha_2 = 0.016$. The learning rates for the brain transfer matrix, cross-stimulus neural mapper, and functional embedder are all set to $1\mathrm{e}{-}5$. We use a batch size of 16 and optimize using Adam. Training on a single NVIDIA A100 GPU achieves convergence in approximately 12 hours.

### 5.2. Dataset

We use the Natural Scenes Dataset (NSD) (Allen et al., 2022), the largest publicly available set, widely used for brain visual decoding. It includes neural responses from subjects viewing complex images from the MSCOCO-2017 dataset(Lin et al., 2014). In line with MindEye2's data-limited setting, our approach uses only a single session of neural recordings, corresponding to one hour of data.

### 5.3. Metrics

To evaluate fMRI-to-image reconstruction performance, we assess both low- and high-level properties of the reconstructed images. Low-level properties capture fundamental visual elements like pixel similarity and edges, while high-level properties reflect semantic information. Following previous works (Scotti et al., 2024a;b), we adopt the PixCorr, SSIM, AlexNet(2), and AlexNet(5) (Krizhevsky et al., 2012)

to evaluate low-level properties and use Inception (Szegedy et al., 2016), CLIP (Radford et al., 2021), EffNet-B (Tan & Le, 2019) and SwAV (Caron et al., 2020) to evaluate high-level properties. These metrics assess the fidelity of the reconstructed images by comparing them with the ground truth. Metric details can be found in Appendix A.

To evaluate functional alignment, we use two metrics: fMRI Spatial Correlation (fSC) (Conroy et al., 2009) and Transfer Quantity (TQ). fSC measures the Pearson correlation between corresponding brain regions of two subjects ($i \neq j$), assessing global alignment consistency. TQ captures voxel-level differences by analyzing the weights of the BTM $\mathcal{M}$, which maps voxels between subjects. For a source voxel indexed by $i$, TQ is defined as $\text{TQ}_i = \sum_{0 \leq j < p} ||\mathcal{M}_{i,j}||$, where $p$ is the number of voxels in the target brain. High TQ values indicate regions with greater activation differences and more intricate functional alignment requirements.

### 5.4. fMRI-based Visual Decoding

We evaluate the visual decoding performance of MindAligner in both qualitative and quantitative manners. The compared state-of-the-art methods include our baseline MindEye2 (Scotti et al., 2024b) and MindBridge (Wang et al., 2024).

**Qualitative Comparison.** We train our model using data from a single session (1 hour of data) and visualize the results in Fig. 3. MindAligner delivers decoding results that are more consistent with the original visual stimuli semantics compared to the baseline, highlighting its effectiveness. The performance improvement can be attributed to the effective learning of brain transfer matrix that accurately aligns the novel subject with the known subject, thus well transferring the pre-trained decoding model to the new subject with limited data.

**Quantitative Comparison.** The first row (1h) of Tab. 1 represents the average of the results from the 2nd to the 5th rows, providing a fair comparison with other methods to demonstrate our superiority. "(subj1) Ours" refers to the average result obtained by aligning the novel subject (subj 1) to every subject in the known subject list (subj 2, 5, 7). Detailed results are provided in Appendix Tab. 7. As summarized in Tab. 1, MindAligner surpasses state-of-the-art methods across almost all metrics. Notably, it achieves a 17.9% improvement in brain retrieval performance. The observed improvement can be attributed to the inherent challenges in implicit alignment strategies employed in previous methods. Aligning multiple subjects with substantial individual differences remains a complex task that may result in information loss during the alignment process. Our method addresses this challenge by adopting an explicit alignment strategy that aligns one subject at a time, avoid conflicts arising from multi-subject alignment. Our model focuses on

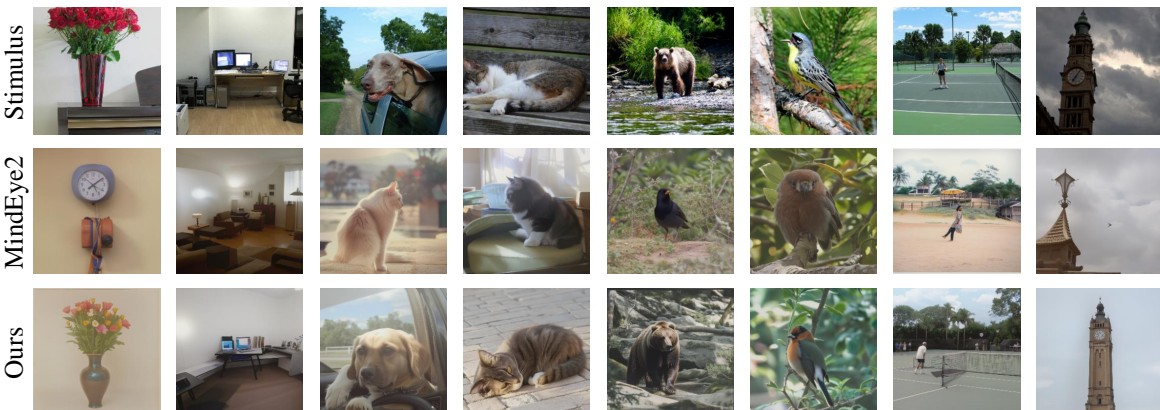

*Figure 3.* Visualization of MindAligner's decoding results from training on one hour of data.

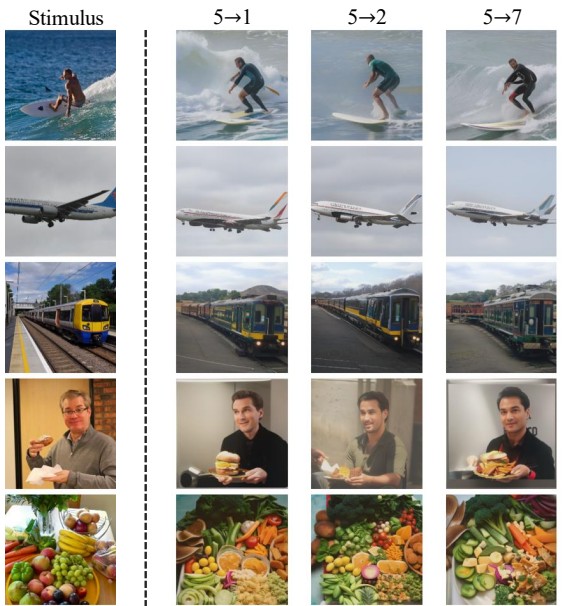

*Figure 4.* Visualization results of aligning a new subject with different known subjects.

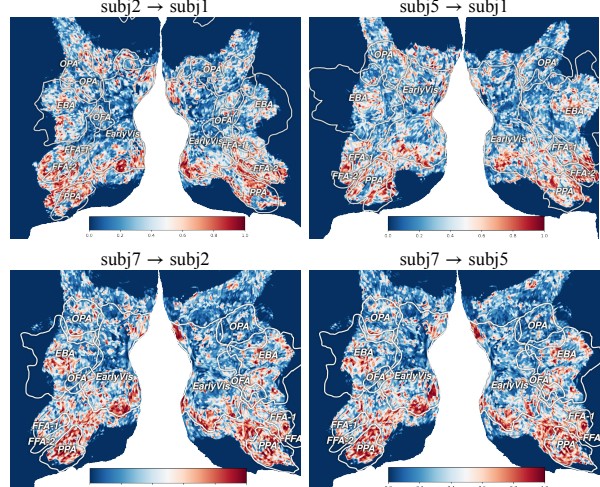

*Figure 5.* Visualization of transfer quantity in brain heatmaps.

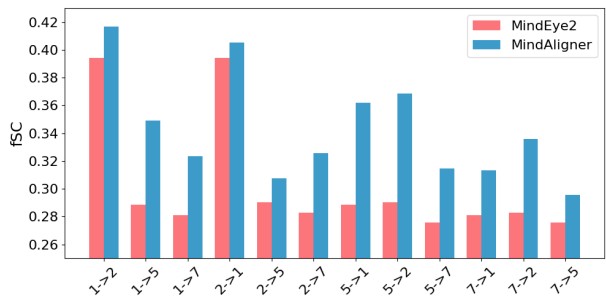

*Figure 6.* Comparison of fSC results between MindAligner and the baseline.

fine-grained cross-subject brain mapping, thereby achieving better decoding performance with high fidelity.

**Ablation Study.** To evaluate the effectiveness of each model design in MindAligner, we perform an ablation study using Subject 2 as the novel subject and Subject 1 as the known subject. The results exclude the refinement step of MindEye2 for generated images. As shown in Tab. 2, training MindAligner with only the visual decoding loss $\mathcal{L}_{dec}$ yields suboptimal cross-subject reconstruction performance, underscoring the difficulty of directly generalizing pre-trained models to new subjects without effective alignment. Adding signal reconstruction loss $\mathcal{L}_{rec}$ significantly enhances performance as it leads to more accurate brain activity reconstructions. The incorporation of $\mathcal{L}_{KL}$ further strengthens alignment by enforcing consistency between the distributions of the generated and real signals. Lastly,

$\mathcal{L}_{latent}$ exploits the correlation of visual stimuli and fMRI embeddings to guide the brain alignment in the latent space, thereby improving model's ability to capture visual semantics in brain activity and enhancing low-level reconstruction performance. These losses together work in synergy to refine alignment and improve cross-subject decoding fidelity.

**Impact of Aligning to Different Subjects.** We visualize the results of fixing a new subject and aligning it with different known subjects in Fig. 4. The reconstruction performance

*Table 2.* Ablation study results. The combination of $\mathcal{L}_{dec}+\mathcal{L}_{rec}+\mathcal{L}_{KL}+\mathcal{L}_{latent}$ is our final model setting.

| Method | PixCorr↑ | SSIM↑ | Alex(2)↑ | Alex(5)↑ | Incep↑ | CLIP↑ | Eff↓ | SwAV↓ |
|---|---|---|---|---|---|---|---|---|
| $+\mathcal{L}_{dec}$ | 0.072 | 0.318 | 63.50% | 71.44% | 66.07% | 62.59% | 0.905 | 0.550 |
| $+\mathcal{L}_{dec}+\mathcal{L}_{rec}$ | 0.186 | 0.340 | 86.83% | 93.51% | 84.55% | 82.42% | 0.811 | 0.465 |
| $+\mathcal{L}_{dec}+\mathcal{L}_{rec}+\mathcal{L}_{KL}$ | 0.191 | 0.407 | 87.98% | 92.99% | **86.61%** | 82.16% | **0.780** | **0.453** |
| $+\mathcal{L}_{dec}+\mathcal{L}_{rec}+\mathcal{L}_{KL}+\mathcal{L}_{latent}$ | **0.195** | **0.408** | **88.25%** | **93.51%** | 86.24% | **82.72%** | 0.782 | 0.454 |

are stable when aligned with different known subjects, as the generated images are nearly identical. This demonstrates the robustness of our brain functional alignment. The choice of the known subject has minimal impact on visual decoding performance. This is because MindAligner leverages the intrinsic correlation between visual semantics to guide the alignment of corresponding brain activities, thereby facilitating robust alignment performance. We provide more detailed cross-subject results in Appendix C.

*Table 3.* Efficiency comparison results. "Tr. Param." refers to the model's trainable parameters when adding a new subject.

| Method | Tr. Param. | Total Param. | Inference |
|---|---|---|---|
| MindEye2 | 2.21G | 2.21G | 5.000 s |
| MindAligner | 139.23M | 2.21G | 5.056 s |

**Computational Efficiency.** We compare the computational efficiency between our model and baseline MindEye2 w.r.t. parameter count and inference time per image. As shown in Tab. 3, MindAligner achieves superior decoding performance while significantly reduces the fine-tuning requirement, with just 6% of MindEye2's learnable parameters, demonstrating its efficiency. Moreover, the addition of our alignment module only slightly increases the inference time.

### 5.5. Brain Functional Alignment Analysis

To deepen the understanding of the brain functional alignment process, we provide detailed visualizations of brain regions along with corresponding functional analysis, offering insights into cross-subject variability and underlying neuroscience mechanisms.

**Region-level Functional Mapping.** We apply the Transfer Quantity (TQ) metric on MindAligner's brain transfer matrix to assess cross-subject associations and visualize the results through brain heatmaps. As shown in Fig. 5, the visualization results highlight two key neuroscience findings. 1) ***The visual system exhibits a hierarchical pattern of inter-subject variability.*** The early visual region (labeled as "EarlyVis" in Fig. 5) presents lower inter-subject variability while higher visual regions (including OPA, FFA, PPA, and EBA) show larger variability. This graded variability aligns with established neuroscientific principles. The early visual region processing fundamental features like lines/textures show more conserved neural mechanisms, sharing larger

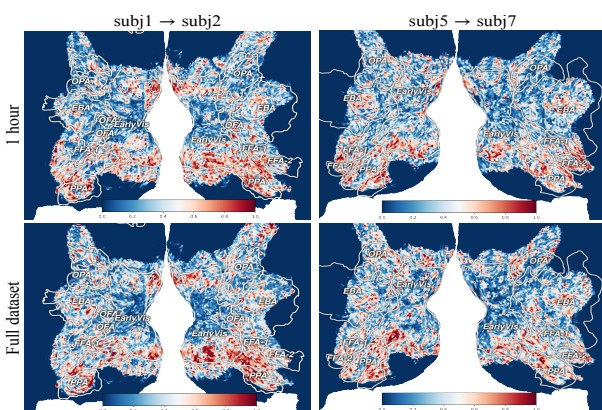

*Figure 7.* Visualization of transfer quantity in brain heatmaps from MindAligner training using 1 hour and full datasets.

commonality across subjects. In contrast, higher visual regions handle more complex cognitive processes, including categorical perception and semantic understanding, leading to higher variability across individuals. 2) ***The ventral pathway exhibits the greatest inter-subject variability.*** The ventral pathway - anatomically positioned on the brain's ventral surface (lower section in Fig. 5) and encompassing functional regions like PPA and FFA - demonstrates the highest variability among visual pathways. This variability arises from its important role in high-level visual processing, such as object recognition, face perception, and semantic interpretation. The ventral stream integrates sensory input with prior knowledge, experiences, and cognitive biases. This results in greater individual differences, as factors like familiarity, attention, and personal experiences shape how visual information is interpreted and understood.

**Cross-subject Correlation Analysis.** To assess the alignment effect of MindAligner, we measure the fMRI Spatial Correlation (fSC) for different subject pairs, comparing our functional alignment strategy with the baseline in reducing inter-subject differences, as shown in Fig. 6. We use $1 \rightarrow 2$ to denote aligning Subject 1 to Subject 2. The results demonstrate that our method significantly outperforms the existing baseline in fSC in all transfer configurations, demonstrating the superiority of our explicit alignment manner against the implicit alignment. By establishing fine-grained voxel correspondences between subjects, MindAligner significantly enhances alignment performance even without paired fMRI signals under shared stimuli, leading to a better visual decoding performance.

**Brain Alignment with More Data.** Furthermore, to evaluate MindAligner's robustness in limited data scenarios, we compare its performance using only 1 hour of fMRI data (2.5% of the full dataset) to using the full scanning sessions. As shown in Fig. 7, even with limited data, the TQ distribution closely resembles that of the full dataset, effectively identifying regions with significant inter-subject variability. This highlights the robustness of our explicit brain alignment strategy under data scarcity.

## 6. Conclusion

We present MindAligner, a functional alignment framework for cross-subject brain visual decoding. Unlike existing methods, it addresses insufficient alignment and lack of interpretability by learning a brain transfer matrix for voxel-level correspondences and proposing a brain functional alignment module for cross-subject mapping. Experiments validate the effectiveness of our method.

## Impact Statement

MindAligner enables high-quality visual perception reconstruction from a single fMRI session, potentially advancing the clinical diagnosis and brain computer interface applications. This approach holds significant potential for enabling alignment across diverse data formats and uncovering commonalities in brain organization across species, such as between humans and monkeys. Moreover, it could play a pivotal role in advancing the creation of unified brain atlases. The datasets used are publicly available, ensuring transparency and participant privacy.

## Acknowledgements

This work was done during the internship at Shanghai Artificial Intelligence Laboratory. This work is supported by Shanghai Artificial Intelligence Laboratory and the NSFC under Grant Nos. U24A20330 and 62361166670.

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

## A. Explanation of Metrics

Following prior work (Scotti et al., 2024a;b), we evaluate the image reconstruction results based on eight metrics, which are categorized into low-level and high-level groups. Low-level metrics, including Pixelwise Correlation (PixCorr), Structural Similarity Index (SSIM) (Wang et al., 2004), AlexNet(2) (Alex(2)), and AlexNet(5) (Alex(5)) (Krizhevsky et al., 2012), focus on textural and structural details. High-level metrics—Inception (Incep) (Szegedy et al., 2016), CLIP (Radford et al., 2021), EfficientNet-B (Eff) (Tan & Le, 2019), and SwAV-ResNet50 (SwAV) (Caron et al., 2020) —assess semantic fidelity. Alex(2), Alex(5), Incep, and CLIP metrics are derived by calculating Pearson correlation between the embeddings of the ground truth and reconstructed images, following the two-way identification framework of Ozcelik and VanRullen (Ozcelik & VanRullen, 2023). Eff and SwAV scores are based on the average distance between feature embeddings.

In addition to the aforementioned metrics, we also evaluate the model using retrieval-based metrics to quantify the fine-grained image information in the fMRI embeddings, following the methodology in MindEye2 (Scotti et al., 2024b). Specifically, for image retrieval, each test fMRI scan is first transformed into its corresponding fMRI representation. We then compute the cosine similarity between this representation and the CLIP-derived image representations of 300 randomly selected images from the test set. Retrieval success is defined as the maximization of cosine similarity between the fMRI embedding and its ground truth CLIP embedding (top-1 retrieval, with random chance at 1/300). To mitigate variability from random batch sampling, the evaluation is repeated 30 times per test sample. The same procedure is applied for brain retrieval, with fMRI and image representations swapped.

## B. Details on Model Parameters

Table 4. Parameter counts of different modules.

| Module | Parameter | Used during inference |
|---|---|---|
| BTM | 122,888,192 | ✓ |
| CNM | 6,299,648 | ✗ |
| FE | 16,781,312 | ✗ |
| MindEye2 | 2,227,290,748 | ✓ |
| MindEye2.ridge_regression | 64,405,504 | ✓ |
| MindEye2.backbone | 1,903,020,028 | ✓ |
| MindEye2.diffusion_prior | 259,865,216 | ✓ |

Table 5. Trainable parameter share of different modules.

| Module | Parameter % | Used during inference |
|---|---|---|
| MindEye2 | 100% | ✓ |
| **MindAligner (Ours)** | 6.2% | - |
| MindAligner.BTM | 5.2% | ✓ |
| MindAligner.CNM | 0.3% | ✗ |
| MindAligner.FE | 0.7% | ✗ |

We list the parameter counts of different modules in the pipeline. MindAligner comprises the Brain Transfer Matrix (BTM), Functional Embedder (FE), and Cross-Stimulus Neural Mapper (CNM). Tab. 4 shows the number of parameters for each module, while Tab. 5 displays the trainalble parameter share for each module, helping us understand their relative contributions to the overall model. The results show that our model has a relatively small parameter count, accounting for only 6% of the parameter size of the visual decoding model. Moreover, the introduction of the FE and CNM modules during the training phase does not significantly increase the model's parameters, contributing to only 1% of the total parameter count. The BTM only accounts for 5% of the parameter size of the visual decoding model.

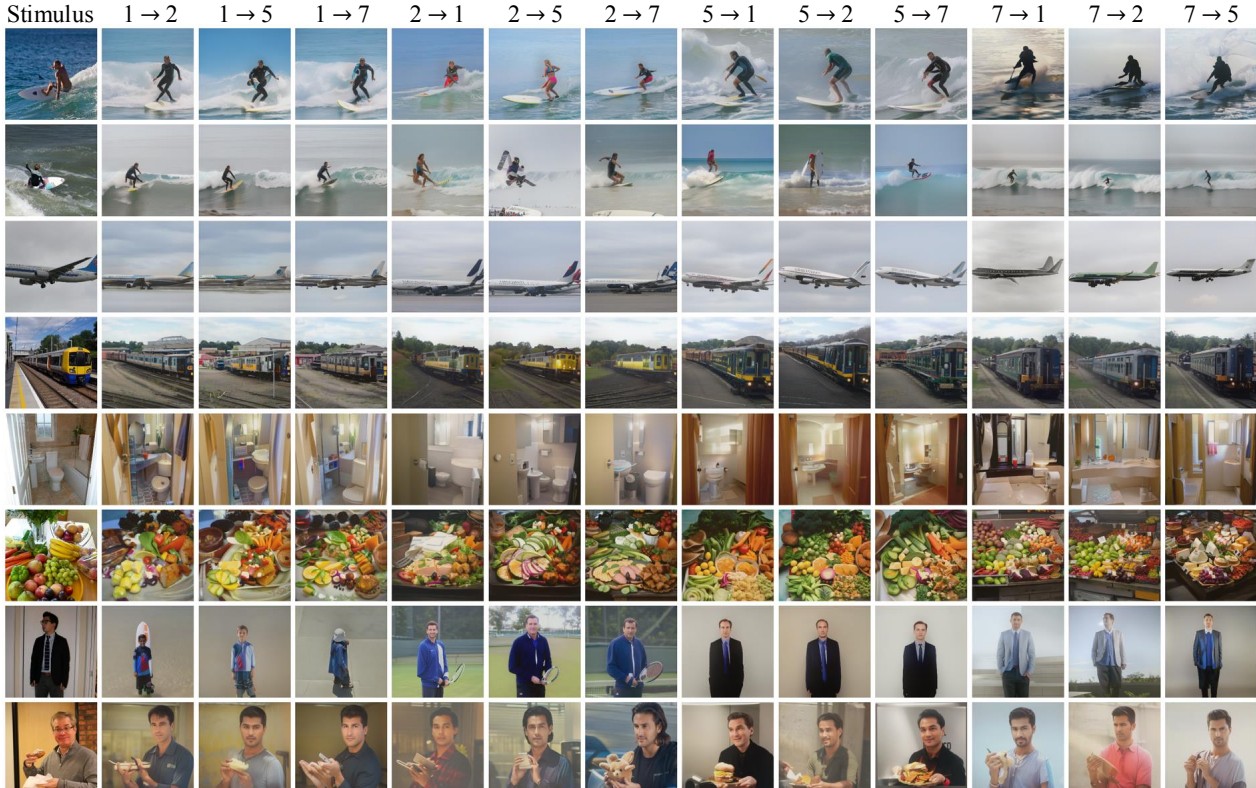

*Figure 8.* More visualization of brain decoding results under different novel and known subjects.

*Table 6.* Performance of MindAligner with different hidden sizes.

| Hidden size | Low-Level | | | | High-Level | | | | Retrieval | |
|---|---|---|---|---|---|---|---|---|---|---|
| | PixCorr↑ | SSIM↑ | Alex(2)↑ | Alex(5)↑ | Incep↑ | CLIP↑ | Eff↓ | SwAV↓ | Image↑ | Brain↑ |
| 64 | 0.144 | 0.384 | 76.33% | 85.34% | 74.67% | 73.99% | 0.876 | 0.512 | 79.13% | 64.00% |
| 256 | 0.166 | 0.395 | 80.83% | 87.81% | 76.89% | 76.47% | 0.858 | 0.498 | 86.67% | 77.83% |
| 512 | 0.185 | 0.405 | 83.85% | 90.95% | 80.74% | 78.55% | 0.839 | 0.481 | 89.06% | 82.97% |
| 1024 | 0.204 | 0.415 | 87.01% | 93.30% | 83.51% | 80.40% | 0.811 | 0.463 | 90.30% | 86.19% |
| 2048 | 0.215 | 0.422 | 88.30% | 93.30% | 83.94% | 82.75% | 0.798 | 0.458 | 90.16% | 85.96% |
| 4096 | 0.218 | 0.425 | 88.36% | 93.55% | 84.17% | 82.57% | 0.794 | 0.455 | 90.09% | 86.19% |

## C. More Results of Aligning to Different Subjects

We visualized more results of fixing a new subject and aligning it with different known subjects in Fig. 8. MindAligner demonstrates robustness, as the generated images remain nearly identical when the novel subject is fixed. This is because MindAligner combines fMRI reconstruction between generated and real data under similar stimuli to ensure result fidelity, while also utilizing intrinsic correlations in visual semantics to guide the alignment of corresponding brain activities, enabling robust optimization.

## D. Ablation Study on Hidden Size

To investigate the potential for further reducing the model size, we adjusted the hidden size and evaluated the model's performance at different values. The experiments showed that when the hidden size is set to 1024, the model delivers comparable performance, while its size is reduced to one-quarter of the original. Compared to the $d = 4096$ configuration, which only accounts for 6% of the framework, the $d = 1024$ setting accounts for just 2%, further highlighting the efficiency

*Table 7.* Detailed performance of our model compared with the baseline. **Bold** means our results outperform the baseline.

| Method | Low-Level | | | | High-Level | | | | Retrieval | |
|---|---|---|---|---|---|---|---|---|---|---|
| | PixCorr↑ | SSIM↑ | Alex(2)↑ | Alex(5)↑ | Incep↑ | CLIP↑ | Eff↓ | SwAV↓ | Image↑ | Brain↑ |
| (subj1) MindEye2 | 0.235 | 0.428 | 88.02% | 93.33% | 83.56% | 80.75% | 0.798 | 0.459 | 93.96% | 77.63% |
| (subj1) Ours | 0.226 | 0.415 | **88.19%** | 93.26% | 83.48% | **81.76%** | 0.800 | **0.459** | 90.90% | **86.88%** |
| subj1→subj2 | 0.222 | 0.413 | **88.09%** | 93.28% | **84.01%** | **81.82%** | 0.796 | **0.457** | 91.56% | **87.49%** |
| subj1→subj5 | 0.227 | 0.416 | **88.29%** | **93.36%** | 83.54% | **80.94%** | 0.803 | 0.461 | 89.76% | **85.78%** |
| subj1→subj7 | 0.229 | 0.416 | **88.18%** | 93.13% | 82.90% | **82.52%** | 0.800 | **0.458** | 91.37% | **87.36%** |
| (subj2) MindEye2 | 0.200 | 0.433 | 85.00% | 92.13% | 81.86% | 79.39% | 0.807 | 0.467 | 90.53% | 67.18% |
| (subj2) Ours | **0.218** | 0.426 | **88.08%** | **93.33%** | **84.13%** | **82.47%** | **0.791** | **0.452** | 90.04% | **85.61%** |
| subj2→subj1 | **0.218** | 0.425 | **88.36%** | **93.55%** | **84.17%** | **82.57%** | **0.794** | **0.455** | 90.09% | **86.19%** |
| subj2→subj5 | **0.218** | 0.426 | **87.88%** | **93.13%** | **83.39%** | **82.05%** | **0.793** | **0.454** | 90.34% | **85.67%** |
| subj2→subj7 | **0.217** | 0.427 | **88.00%** | **93.32%** | **84.83%** | **82.78%** | **0.785** | **0.449** | 89.70% | **84.98%** |
| (subj5) MindEye2 | 0.175 | 0.405 | 83.11% | 91.00% | 84.33% | 82.53% | 0.781 | 0.444 | 66.94% | 46.96% |
| (subj5) Ours | **0.197** | **0.409** | **84.69%** | **91.61%** | **84.63%** | **82.76%** | 0.784 | 0.454 | **70.62%** | **65.95%** |
| subj5→subj1 | **0.196** | 0.405 | **84.23%** | **91.28%** | **84.66%** | **82.93%** | 0.787 | 0.459 | **69.67%** | **65.14%** |
| subj5→subj2 | **0.196** | **0.409** | **84.71%** | **91.88%** | **84.56%** | **82.88%** | 0.783 | 0.455 | **70.78%** | **66.38%** |
| subj5→subj7 | **0.198** | **0.412** | **85.12%** | **91.67%** | **84.66%** | 82.47% | **0.781** | 0.450 | **71.41%** | **66.32%** |
| (subj7) MindEye2 | 0.170 | 0.408 | 80.70% | 85.90% | 74.90% | 74.29% | 0.854 | 0.504 | 64.44% | 37.77% |
| (subj7) Ours | **0.183** | 0.407 | **81.45%** | **88.31%** | **79.92%** | **77.82%** | **0.834** | **0.487** | 64.18% | **62.58%** |
| subj7→subj1 | **0.180** | 0.404 | **80.86%** | **87.47%** | **78.94%** | **77.05%** | **0.840** | **0.492** | 65.62% | **63.69%** |
| subj7→subj2 | **0.185** | 0.406 | **82.11%** | **89.01%** | **80.47%** | **77.53%** | **0.835** | **0.486** | 63.26% | **61.06%** |
| subj7→subj5 | **0.183** | **0.411** | **81.38%** | **88.46%** | **80.36%** | **78.87%** | **0.828** | **0.482** | 63.67% | **62.98%** |
| (1 h) MindEye2 | 0.195 | 0.419 | 84.2% | 90.6% | 81.2% | 79.2% | 0.810 | 0.468 | 79.0% | 57.4% |
| (1 h) Ours | **0.206** | 0.414 | **85.6%** | **91.6%** | **83.0%** | **81.2%** | **0.802** | **0.463** | 78.9% | **75.3%** |

performance advantages of our model.

## E. More Detialed MindAligner Reconstruction Performance

We provide more detailed MindAligner reconstruction evaluation results, as shown in Tab. 7. MindAligner surpasses the baseline in almost all metrics, even when applying the same novel subject to different known subjects. Notably, our method achieves a 17.9% improvement in brain retrieval performance. This improvement stems from addressing the limitations of implicit alignment strategies used in prior methods. Aligning multiple subjects with significant individual differences is inherently challenging and often leads to functional information loss during the alignment process. To overcome this, our approach employs an explicit alignment strategy, aligning one subject at a time, which effectively mitigates the conflicts arising from multi-subject alignment. By focusing on region-level cross-subject brain mapping, our model not only achieves superior visual performance but also captures more comprehensive brain region features for functional representation.

## F. The Linear Hypothesis Underlying MindAligner

*Table 8.* Comparison of different methods across various metrics.

| Method | PixCorr↑ | SSIM↑ | Alex(2)↑ | Alex(5)↑ | Incep↑ | CLIP↑ | Eff↓ | SwAV↓ |
|---|---|---|---|---|---|---|---|---|
| FE (Transformer) | 0.182 | 0.350 | 87.68% | 93.55% | 85.14% | 81.65% | 0.807 | 0.463 |
| NM (Transformer) | 0.169 | 0.339 | 85.40% | 92.46% | 84.23% | 80.85% | 0.819 | 0.482 |
| Ours (Linear) | **0.195** | **0.408** | **88.25%** | 93.51% | **86.24%** | **82.72%** | **0.782** | **0.454** |

The linear hypothesis in cross-subject brain difference modeling is a well-established principle in neuroscience. Haxby et al. (2011) demonstrated that inter-subject differences in visual representations can be eliminated via linear transformations, while Naselaris et al. (2011) showed that linear models account for over 90% of variance in primary sensory cortex decoding. Drawing on these established linear hypotheses, MindAligner employs linear structures to model brain variations across subjects and stimuli.

To further substantiate the validity of our hypothesis, we perform ablation experiments on the subj2→subj1 transfer setting by replacing the original Functional Embedder (FE) and Cross-stimulus Neural Mapper (NM) components with nonlinear architectures, specifically employing Transformer layers. This allows us to directly assess the contribution of the linear design in each module and evaluate the impact of introducing nonlinearity on model performance and generalization.

The superior performance of the linear layer further validates the linear hypothesis, demonstrating its effectiveness in modeling individual brain differences in limited data settings.

