# OpenReview forum: "MindAligner: Explicit Brain Functional Alignment for Cross-Subject Visual Decoding from Limited fMRI Data"
_ICML.cc/2025/Conference — ICML 2025 poster_

### Official Review · Reviewer_Hn6R · 2025-02-26

**Overall Recommendation:** 2

**Summary:**

This paper proposes MindAligner, a framework using functional alignment to facilitate cross-subject brain decoding. Their framework consists of (1) “Brain Transfer Matrix” (BTM) that works by transforming limited novel subject brain activity into the brain activity of a known, previously seen subject via linear mappings, such that the researcher can use a pretrained model learned on someone else’s brain rather than training a new model from scratch on the new subject’s brain, and (2) a “Brain Functional Alignment” module facilitates BTM learning by estimating different alignment losses using similar images seen by both participants.

**Claims And Evidence:**

Claim 1: MindAligner outperforms existing methods in fMRI-to-image reconstruction in low-sample settings.

Evidence: Table 1 compares MindAligner results to past work (MindEye2 and MindBridge) using 1 hour of data from a new subject. They show MindAligner performs best across most metrics. The authors don’t seem to reveal which “known” subject was used for the results depicted in this table. They also do not explicitly mention that the top header is the average of the subsequent four rows (averaging across the 4 subjects’ metrics), nor do they clarify that the subsequent four rows were conducted with only 1 hour of training data for the subject in parenthesis. These are all things that should be easily correctable in a revision.

Claim 2: MindAligner facilitates novel insights in cross-subject functional analysis.

Evidence: The paper claims their method allows for novel brain functional alignment analyses, allowing for improved interpretations that past methods lack. They present two such functional alignment results: (1) Region-level Functional Mapping and (2) Cross-subject Correlation Analysis.

Region-level Functional Mapping: This result is simply presenting which regions of the brain had the greatest inter-subject variability. You could achieve such results by simply comparing the input voxels across subjects, without any model training needed. What unique advantages does MindAligner offer to region-level functional mapping that statistical comparison across normalized flatmaps of the participants can’t already do?

Cross-subject Correlation Analysis: The authors compute the functional correlation between brain regions of two subjects. I don’t see what novel insights can be gleaned from this analysis? The authors simply state that these results outperform MindEye2, but that doesn’t relate to the question of novel insights. Further, I do not understand how the “baseline” of MindEye2 was computed? MindEye2 doesn’t involve translating from one subject’s voxel space to another subject’s voxel space, so what is even being correlated?

Claim 3: “MindAligner [is] the first explicit brain alignment framework that enables cross-subject visual decoding and brain functional analysis in the data-limited setting.”

MindAligner is not the first to tackle cross-subject decoding by explicitly mapping a new subject’s brain to an old, known subject’s brain. Specifically, Ferrante et al. (2024) seems nearly identical to the method used by the present authors except for different losses implemented in the Brain Functional Alignment module. Further, Ferrante et al. (2024) was also specifically used within the context of wanting to get decoding models to work with smaller datasets.

I’m also unsure if the term “explicit” is the best way to describe their method in contrast to previous work using shared-subject spaces. Past methods mapped new subjects to a shared subject space, but in practice these past methods could just as easily have made the “shared subject space” consist of a single subject rather than multiple subjects (in fact this is one of the ablations in the MindEye2 paper, in Appendix A.11).

My understanding of the novelty here seems to be that mapping to a single subject is more advantageous (in terms of decoding performance) than mapping to multiple subjects and that the use of BTM confers novel interpretational insights (except see my concerns regarding this in Claim 2 section above). Calling this method “explicit” whereas past works are “implicit” seems confusing—why is it that mapping from one subject to another is explicit but mapping from one subject to several subjects is implicit? Would you call the works of Bazeille et al., 2019, Thual et al., 2022; 2023, and Ferrante et al., 2024 explicit or implicit approaches?

Overall I think only claim #1 is sufficiently supported out of the above three claims, and even then, the novelties of their method seem largely borrowed from Ferrante et al. (2024).

**Essential References Not Discussed:**

None

**Experimental Designs Or Analyses:**

I find it misleading that Figure 1 depicts the authors approach as mapping different functional regions of a new subject’s brain to various functional “atlas” in a known subject. Neuroimagers use the term “atlas” to mean a predefined region-of-interest or parcellation scheme (which could confer additional interpretational benefits). However, the use of an “atlas” or “parcel” is not used anywhere else in the paper. The method section seems to suggest that it’s simply linear mappings from one voxel space to another, with no atlas involved, making the Figure 1 potentially misleading.

**Methods And Evaluation Criteria:**

The proposed methods and evaluation criteria make sense, although further clarifications are recommended (see other comments).

**Other Comments Or Suggestions:**

The authors should re-examine their tables for proper bolding of the best performing metrics. Some metrics are misbolded and other metrics that should be bolded are not.

“However, achieving such brain alignment is challenging, as it requires paired fMRI from subjects performing the same task (i.e., viewing identical visual stimuli (Bazeille et al., 2021)), a condition not met by the existing dataset (Allen et al., 2022).”
Note that Allen et al. (2022) did contain a subset of images seen by all subjects. The authors choose not to use this in favor of using it as the test set.

**Other Strengths And Weaknesses:**

Methods novelty is limited and claims regarding novel interpretational benefits seem not well-justified. That said, it's still useful and important to show that simple linear alignment to someone else's voxel space can outperform the more complicated shared-subject latent space approaches used in other work.

**Questions For Authors:**

I could not find what the authors used for their “pre-trained brain decoding model”. Was it MindEye2 without shared-subject pretraining? Or something else? If it was using MindEye2 pretrained with a shared-subject latent space then that would make interpretations and comparisons with MindEye2 confusing (you'd basically be benefitting from both "explicit" *and* "implicit" advantages from other subjects).

Was the same brain region of interest (nsdgeneral) used here as other works like MindEye?

**Relation To Broader Scientific Literature:**

Broader scientific literature is well-covered, with exception to my points regarding Ferrante et al. (2024) (see Claims And Evidence)

**Theoretical Claims:**

See my Claims And Evidence section comments.

---

> ### Author Rebuttal · Authors · 2025-04-01
>
> Thank you for your time and valuable feedback. We will incorporate the suggested modifications in the revised version.
>
> **Q1: Clarification on "explicit" and "implicit" functional alignment**
>
> We define alignment in voxel space as explicit and methods in latent space that cannot be restored to voxel space as implicit, not based on single- or multi-subject distinction. We distinguish these two to highlight their implications for cross-subject functional interpretability.
>
> **Q2: Comparison with Ferrante et al. (2024) using Simple Alignment Techniques (SAT).**
>
> **Novelty**: Both MindAligner and SAT use linear hypothesis of cross-subject variances (see Reviewer 5B7Z Q2), but their motivations differ, and linear modeling is **NOT** MindAligner's primary novelty. SAT needs identical stimuli across subjects-a strict requirement for practical use. Conversely, MindAligner tackles alignment without identical stimuli, a critical issue also emphasized by Wang et al., 2024. We address this issue to perform soft cross-subject brain alignment under different stimuli. Thus, MindAligner is the first to achieve explicit functional alignment **WITHOUT** requiring identical stimuli.
>
> **Performance comparison**:
> We adapt SAT to our setting for comparison, and the results are in Tab.2: https://mindaligner.github.io/MindAligner/. SAT significantly underperforms MindAligner, showing that simple linear alignment **ALONE** cannot address the lack of shared stimuli like MindAligner does.
>
> **Q3: NSD had few images seen by all subjects, which the authors chose to use as the test set.**
>
> We use the standard train/test splits from the original NSD dataset and previous NSD reconstruction papers, including MindEye2, to ensure a fair comparison.
>
> **Q4: About the setting of Table 1.**
>
> Please see the Reviewer iSeR Q1.
>
> **Q5: Comparison between MindAligner and normalized flatmaps in region-level functional mapping.**
>
> Compared to normalized flatmaps, MindAligner offers several advantages:
> 1. **Pattern analysis without paired stimuli data**: By training on a small amount of data, MindAligner could capture fine-grained neural patterns without paired stimuli, rather than regional average activations.
> 2. **Region-level analysis**: MindAligner could evaluate the contribution of brain regions to specific analysis, whereas simple statistical comparisons of flatmaps without training can only identify correlations. Compared to flatmaps, MindAligner achieves effective brain region mapping and enables exploration of region-level correspondence between arbitrary subjects, as shown in Reviewer iSeR.Q4.
>
> **Q6: About the Cross-subject Correlation Analysis (CCA).**
>
> CCA reveals two key insights:
> 1. CCA results show that MindAligner's explicit alignment manner achieves higher brain correlation scores than implicit alignment. Benefit from effective subject alignment, MindAligner achieves superior decoding performance.
> 2. We further analyze the relationship between alignment and decoding, finding that subjects with higher average correlation scores (e.g., subj1 and subj2) with other subjects consistently achieve better visual decoding performance. Please refer to Tab.3: https://mindaligner.github.io/MindAligner/.
>
> Together, these findings confirm that improving cross-subject brain alignment is an effective and direct way to enhance novel subject decoding, offering a pathway for future research on cross-subject decoding.
>
> **Q7: About the pre-trained brain decoding model.**
>
> To ensure fairness, we follow MindEye2 to use its multi-subject pre-trained weights and learn BTM for novel subjects under limited data. We will clarify it in the revised version. It is worth noting that the alignment layer (i.e., Linear Regression (LR)) in multi-subject MindEye2 only aligns known subjects to shared space and can not be directly applied to new subjects without fine-tuning. The results of MindEye2 in Tab.1 is its multi-subject pre-trained model (same as MindAligner) with its LR fine-tuned on novel subjects.
> In contrast, MindAligner does not involve this process, thus does **NOT** enjoy the benefit of implicit alignment.
>
> **Q8: How the "baseline" results of MindEye2 in Fig. 6 is computed?**
>
> For MindEye2, we input paired stimuli in test set into each subject’s ridge regression model and compute the fSC between embeddings. For MindAligner, we transform the novel subject’s voxels via BTM, reconstruct the known subject’s voxels, and compute the fSC between the reconstructed and paired stimuli fMRI after the linear regression in multi-subject MindEye2.
>
>
> **Q9: About the term "atlas".**
>
> Thank you for your suggestion. In our paper, we used "atlas" to describe functional regions during brain alignment process. To avoid misunderstanding, we will revise it to "region".
>
> **Q10: About the brain ROI?**
>
> We strictly followed the same ROI selection protocol as other works for fair comparison.
>
> **Q11: Misbolded results.**
>
> We will revise them.

---

### Official Review · Reviewer_7q81 · 2025-03-11

**Overall Recommendation:** 3

**Summary:**

This work considers inter-subject alignment in the context of image reconstruction from fMRI. A light way Brain Transfer Matrix is used to align a novel subject to a known subject, which differs from the previous work that aligns subjects in a shared latent space.

**Claims And Evidence:**

Most claims are well-stated and supported by evidence and references. I have one concern though.

In Table 3, the Tr. Param. of MindEye2 is 2.21G (basically the total param). From my understanding, adding a new subject only requires training a linear layer in MindEye2. Am I missing something here?

**Essential References Not Discussed:**

N/A

**Experimental Designs Or Analyses:**

The experimental designs and analyses are reasonable, except that some results in Table 1 are very close, and a statistical test could be a better choice to support the claims.

**Methods And Evaluation Criteria:**

The evaluation criteria make sense. But I do have a question (concern) about the method.

How does this method compare with finetuning a decoding model (say mind-eye) with LoRa using the novel subject? It would still be efficient in terms of training and avoid the trouble of finding a suitable subject to align.  What does aligning to a specific subject bring us compared with that?

**Other Comments Or Suggestions:**

N/A

**Other Strengths And Weaknesses:**

This paper is well-written and structured. The major concerns are stated in the previous parts.

**Questions For Authors:**

Questions are stated in the previous parts.

**Relation To Broader Scientific Literature:**

Cross-subject adaptation has been an important and interesting problem in computational neuroscience tasks. Cross-subject capability has been a limiting factor in many attempts to make algorithms practical in this field.

**Theoretical Claims:**

No theoretical claims in this work.

---

> ### Author Rebuttal · Authors · 2025-04-01
>
> Thank you for your constructive feedback. We will address your concerns point by point below:
>
> **Q1: About the training parameters of MindEye2.**
>
> According to the open-source code of MindEye2, it performs full-parameter fine-tuning when adding a new subject. The results in Table 3 are obtained by adhering precisely to this methodology. We also evaluate the performance of MindEye2 by fine-tuning only its linear layer when adding a new subject (Finetune_ridge (avg)):
>
> | Method               | PixCorr↑  | SSIM↑     | Alex(2)↑  | Alex(5)↑  | Incep↑    | CLIP↑     | Eff↓      | SwAV↓     | Image↑    | Brain↑    |
> | -------------------- | --------- | --------- | --------- | --------- | --------- | --------- | --------- | --------- | --------- | --------- |
> | Finetune_ridge (avg) | 0.172     | 0.360     | 80.6%     | 87.8%     | 79.8%     | 78.9%     | 0.830     | 0.486     | 79.0%     | 55.2%     |
> | Ours (avg) | **0.206** | **0.414** | **85.6%** | **91.6%** | **83.0%** | **81.2%** | **0.802** | **0.463** | **79.0%** | **75.3%** |
>
> The results demonstrate that fine-tuning only the ridge layer is still less effective compared to MindAligner.
>
> **Q2: How does this compare to LoRA fine-tuning? What are the benefits of subject-specific alignment?**
>
> Compared to LoRA, our method enjoys the following advantages:
> 1. **Enhanced decoding performance**: we conduct ablation experiments to use LoRA finetuning for subj 2→1 alignment. The results show that LoRA underperforms compared to MindAligner, particularly in PixCorr and SSIM, indicating that MindAligner better preserves the low-level information of the visual stimulus.
>
> | Method               | PixCorr↑  | SSIM↑ | Alex(2)↑   | Alex(5)↑   | Incep↑     | CLIP↑      | Eff↓      | SwAV↓     |
> | -------------------- | --------- | ----- | ---------- | ---------- | ---------- | ---------- | --------- | --------- |
> | LoRA (on subj2)      | 0.175     | 0.363 | 85.75%     | 92.72%     | 85.26%     | 80.31%     | 0.812     | 0.459     |
> | Ours subj2->1        | **0.195** | 0.408 | 88.25%     | 93.51%     | 86.24%     | 82.72%     | 0.782     | 0.454     |
> | Ours subj2->1 + LoRA | 0.193     | 0.408 | **88.28%** | **93.81%** | **86.51%** | **83.11%** | **0.776** | **0.450** |
>
> 2. **Neuroscience Interpretability**: MindAligner can reveal fine-grained functional correspondences that enhance neuroscientific interpretability, a capability that simply using LoRA fails to achieve.
>
> **Q3: Some results in Tab.1 are close, and a statistical test could be a better choice to support the claims.**
>
> Thank you for the suggestion. We have conducted Wilcoxon signed-rank tests comparing our method with MindEye2. The results are summarized in the table below, with significant improvements (p < 0.05) highlighted in bold.
>
> | Setting           | Avg       | Subj1 | Subj2     | Subj5     | Subj7     |
> | ----------------- | --------- | ----- | --------- | --------- | --------- |
> | Ours vs. MindEye2 | **0.008** | 0.703 | **0.005** | **0.032** | **0.005** |
>
> Here, "Subj1" represents the average result of our method when Subj1 is used as the novel subject mapped to Subj2, 5 and 7, compared to MindEye2's results on Subj1. "Avg" refers to the average performance difference between our method and MindEye2 across all subjects. The results confirm statistically significant improvements in our approach in most scenarios. Updated analyses will be included in the revised version.

---

### Official Review · Reviewer_iSeR · 2025-03-13

**Overall Recommendation:** 2

**Summary:**

The manuscript proposed an explicit brain functional alignment for cross-subject decoding. The method trains a cross-subject brain transfer matrix to map signals from novel subjects to the known subject. Experimental results demonstrate improved performance and provide insightful interpretations.

## update after rebuttal

The authors have addressed most of my initial concerns. However, I still believe the proposed method does not demonstrate sufficient technical contributions. The core contribution appears to be learning a linear mapping to calibrate voxels between a new subject and existing subjects, which I find relatively naive. For a machine learning venue such as ICML, I expect application-specific models to incorporate nontrivial methodological or algorithmic advancements, motivated by domain-specific insights. In my view, this manuscript does not demonstrate a clear advantage in that regard. Therefore, I have decided to retain my original score.

**Claims And Evidence:**

The settings of the main table are unclear to me. Therefore, it is hard to say if the claim holds (see questions below in my review) without further clarifications from the authors.

**Essential References Not Discussed:**

[1] works on cross-subject decoding, but is not discussed.

[1] Wang, Zicheng, et al. "UniBrain: A Unified Model for Cross-Subject Brain Decoding." arXiv preprint arXiv:2412.19487 (2024).

**Experimental Designs Or Analyses:**

The experiments are reasonable. The ablation study is sufficient regarding the proposed components. The analysis is interesting. However, visualizing and analyzing the transferability of specific voxels would be more interesting. For example, something like "FFA1 shifts to the right when transferring from subject 1 to subject 2" would be more significant.

**Methods And Evaluation Criteria:**

The method makes sense and makes nontrivial contributions in terms of the explicit modeling of transferability between subjects. However, some existing solutions that align two distributions should be considered first (e.g. optimal transport) and should be included as baselines. The evaluation makes sense. However, some other baselines are still missing (e.g., [1]).

[1] Wang, Zicheng, et al. "UniBrain: A Unified Model for Cross-Subject Brain Decoding." arXiv preprint arXiv:2412.19487 (2024).

**Other Comments Or Suggestions:**

1. In section 4, I would recommend annotating the dimension of each notation (e.g. $\mathcal{F}_N$)
2. It is unusual to me to use a mathcal symbol as a notation for a matrix. Consider changing them to plain upper letters or referring somewhere else that uses similar notations.

**Other Strengths And Weaknesses:**

Strength:
The work enables cross-subject fMRI alignment with interpretations.

Weakness:
I did not find any major weaknesses other than what have been discussed elsewhere.

**Questions For Authors:**

1. The setting of the main table (table 1) is very unclear. Did you train your model on each subject individually? Or did you set each subject as the novel subject? If it is the latter, which subject did you set as the known subject? You have that information in ablation studies, but I did not find it in table 1.
2. In table 1, why does MindBridge only appear in subject 1, but not subject 2, 5, 7?

**Relation To Broader Scientific Literature:**

This could be related to cross-subjects fMRI foundation models, as well as multimodal (besides images, e.g., text) brain decoding.

**Theoretical Claims:**

I did not find any issues regarding theoretical claims.

---

> ### Author Rebuttal · Authors · 2025-04-01
>
> Thank you for your valuable time. We will revise the typos as suggested. **We promise to open-source our code upon acceptance.**
>
> **Q1: The settings of the Tab. 1 is unclear.**
>
> Thank you for your question. The setting in Tab. 1 is as follows: "Ours (subj 1)" refers to the average result obtained by aligning the novel subject (subj 1) to every subject in the known subject list (subj 2, 5, 7).
> Detailed results are provided in Appendix Tab. 7, where "1 → 2" denotes the experiment with subj 1 as the novel subject and subj 2 as the known subject. We will clarify the setting in the revised version.
>
> **Q2: More results of MindBridge**
>
> We provide detailed MindBridge results for each subject in Tab. 1 in the anonymous link (https://mindaligner.github.io/MindAligner/). Our method outperforms MindBridge in each subject, demonstrating the effectiveness of MindAligner.
>
> **Q3: Comparison with UniBrain**
>
> 1. **Method and quantitative performance comparison**: we reimplement UniBrain using their official code and following the same setting in MindAligner in the table below. Complete results on subjects can refer to Tab. 1 in the anonymous link (https://mindaligner.github.io/MindAligner/)
>
> | Method         | PixCorr↑  | SSIM↑     | Alex(2)↑  | Alex(5)↑  | Incep↑    | CLIP↑     | Eff↓      | SwAV↓     |
> | -------------- | --------- | --------- | --------- | --------- | --------- | --------- | --------- | --------- |
> | Unibrain (avg) | 0.078     | 0.222     | 74.2%     | 82.6%     | 75.9%     | 80.1%     | 0.865     | 0.542     |
> | Ours (avg)     | **0.206** | **0.414** | **85.6%** | **91.6%** | **83.0%** | **81.2%** | **0.802** | **0.463** |
>
> The results indicate that MindAligner demonstrates superior performance compared to Unibrain, especially in low-level metrics.
> UniBrain attempts multi-subject decoding with shared parameters by aligning semantics in the latent space. However, this implicit alignment approach may suffer from severe semantic conflicts, leading to suboptimal results.
> In contrast, MindAligner leverages explicit alignment, preserving semantic features while avoiding semantic conflicts, thereby achieving superior performance.
> 2. **Qualitative Comparison**:
> The comparison of reconstruction results are available in Fig.1 on the anonymous link(https://mindaligner.github.io/MindAligner/).
> The results indicate that compared to UniBrain, MindAligner achieves superior semantic preservation in the generated images, which align closely with the visual stimuli.
>
> **Q4: Some existing solutions that align two distributions should be considered first (e.g. optimal transport (**OT**)) and should be included as baselines.**
>
> Thank you for your suggestion. We follow your suggestion to include results of using OT to fit the KL distribution of fMRI data, the results are as follows:
>
> | Method     | PixCorr↑  | SSIM↑     | Alex(2)↑  | Alex(5)↑  | Incep↑    | CLIP↑     | Eff↓      | SwAV↓     |
> | ---------- | --------- | --------- | --------- | --------- | --------- | --------- | --------- | --------- |
> | OT         | 0.107     | 0.201     | 77.3%     | 81.6%     | 70.5%     | 71.8%     | 0.890     | 0.534     |
> | Ours (avg) | **0.206** | **0.414** | **85.6%** | **91.6%** | **83.0%** | **81.2%** | **0.802** | **0.463** |
>
> Our model demonstrates superiority over OT method. Without shared stimulus-fMRI paris across subjects, OT methods struggle to produce effective brain alignment. MindAligner addresses this limitation through cross-stimulus mapping and multi-level alignment loss.
>
> **Q5: It would be more insightful to visualize and analyze the transferability of specific voxels.**
>
> We visualize the transferability of the early visual cortex V2 region as a representative example. Specifically, for each voxel in the novel subject, we identify the corresponding voxel in the known subject with the highest correspondence weight in the BTM and visualize the results in Fig. 2: https://mindaligner.github.io/MindAligner/.
> Due to large cross-subject variances, the brain region exhibits location variations across novel and known subjects, and from Fig. 2, we can observe how specific region’s activity patterns map across subjects.
> Interestingly, the mapped V2 regions for novel subjects align accurately with their actual anatomical locations, e.g., visualization in "novel subj1" v.s. "novel subj2 → known subj1".
>
> **Q6: Notation issues.** We will revise them.

---

> > ### Comment · Reviewer_iSeR · 2025-04-03
> >
> > Thanks for the reply. Now I understand the setting of Table 1's 2nd-5th rows, but I am still confused about the setting of the first row. Is it training the model on all subjects with 1hr data?

---

> > > ### Author Response · Authors · 2025-04-04
> > >
> > > Thanks for your kind reply and valuable time. The first row (1h) actually represents the average of the results from the 2nd to the 5th rows, providing a fair comparison with other methods to demonstrate our superiority. We will incorporate the explanations in the revised version. Once again, we sincerely appreciate your insightful feedback, and we hope our response has fully resolved your concerns. We would greatly appreciate it if you could kindly consider raising your rating.

---

### Official Review · Reviewer_5B7Z · 2025-03-14

**Overall Recommendation:** 4

**Summary:**

This paper propose MindAligner, a explicit brain signal functional alignment framework for cross-subject brain decoding. It utilize a LoRA-based **brain transfer matrix (BTM)** to convert signals from novel subjects into signals fo a known subjects. A **brain functional alignment (BFA)** module based on linear hypothesis is designed to acomplish this. Experimental results shows superior performance compared to existing methods.

**Claims And Evidence:**

Yes

**Essential References Not Discussed:**

.

**Experimental Designs Or Analyses:**

Yes

**Methods And Evaluation Criteria:**

Yes

**Other Comments Or Suggestions:**

The term "multi-level" typically implies more than two levels. If only two levels are considered, a term like "bi-level" might be more appropriate.

**Other Strengths And Weaknesses:**

## Strengths

1. The motivation is strong and well-founded, as the identified limitations of current cross-subject methods are valuable to address.
2. The introduction of explicit soft alignment is novel and promising, particularly for handling non-common views between known and novel subjects.
3. The interpretable functional alignment analysis is good, providing valuable insights into cross-subject variability and the underlying neural mechanisms.


## Weaknesses

1. Many technical components (e.g., linear modulation, functional embedder) rely on the linear hypothesis, yet this hypothesis is neither explicitly presented nor justified, making the paper less convincing in terms of theoretical foundation.
3. The clarity of descriptions needs improvement, as some key aspects are missing, which hinders the reader's ability to grasp the full scope of the work. For example:
* What distance metric is used to measure dissimilarity?
* Why is MindBridge only compared on a single entry? Additionally, why is there no visual comparison with MindBridge?
4. The improvement introduced by the proposed latent alignment loss is minor, based on the ablation study in Table 2.

**Questions For Authors:**

Please refer to the weaknesses section.

If these concerns are clearly addressed, I will consider raising my score.

**Relation To Broader Scientific Literature:**

Previous cross-subject brain decoding works primarily rely on implicit alignment, whereas this work proposes explicit alignment, introducing a new perspective in the field.

**Theoretical Claims:**

Yes

---

> ### Author Rebuttal · Authors · 2025-04-01
>
> We sincerely appreciate your valuable time and insightful feedback. Your recognition is highly meaningful to us. Below is our response addressing your concerns：
>
> **Q1: Multi-level -> Bi-level.**
>
> Thank you for your suggestion. We will revise it as suggested.
>
> **Q2: About the linear hypothesis in BFA.**
>
> **Literature-based Justification**: The linear hypothesis in cross-subject brain difference modeling is a well-established principle in neuroscience. Haxby et al. (2011) demonstrated that inter-subject differences in visual representations can be eliminated via linear transformations, while Naselaris et al. (2011) showed that linear models account for over 90% of variance in primary sensory cortex decoding. Drawing on these established linear hypotheses, MindAligner employs linear structures to model brain variations across subjects and stimuli.
>
> **Experimental Justification**: To further validate this, we conduct **ablation experiments on subj2->1** by replacing the Functional Embedder (FE) and Cross-stimulus Neural Mapper (NM) structures with nonlinear architectures (Transformer layer).
>
> | Method           | PixCorr↑  | SSIM↑     | Alex(2)↑   | Alex(5)↑   | Incep↑     | CLIP↑      | Eff↓      | SwAV↓     |
> | ---------------- | --------- | --------- | ---------- | ---------- | ---------- | ---------- | --------- | --------- |
> | FE (Transformer) | 0.182     | 0.350     | 87.68%     | **93.55**% | 85.14%     | 81.65%     | 0.807     | 0.463     |
> | NM (Transformer)  | 0.169     | 0.339     | 85.40%     | 92.46%     | 84.23%     | 80.85%     | 0.819     | 0.482     |
> | Ours (Linear)    | **0.195** | **0.408** | **88.25%** | 93.51%     | **86.24%** | **82.72%** | **0.782** | **0.454** |
>
> The superior performance of the linear layer further validates the linear hypothesis, demonstrating its effectiveness in modeling individual brain differences in limited data settings.
>
> **Q3: Distance metric.**
>
> Cosine similarity was employed as the distance metric.
>
> **Q4: About the improvement of Latent Alignment loss (LALoss).**
>
> To thoroughly investigate the impact of LALoss, we conduct additional ablation experiments on subj 2→1 by only adding LALoss to the baseline visual decoding loss（DecLoss）:
>
> | Method           | PixCorr↑  | SSIM↑     | Alex(2)↑   | Alex(5)↑   | Incep↑     | CLIP↑      | Eff↓      | SwAV↓     |
> | ---------------- | --------- | --------- | ---------- | ---------- | ---------- | ---------- | --------- | --------- |
> | DecLoss          | 0.072     | 0.318     | 63.50%     | 71.44%     | 63.07%     | 62.59%     | 0.935     | 0.550     |
> | DecLoss + LALoss | **0.187** | **0.348** | **87.92%** | **92.19%** | **84.47%** | **82.78%** | **0.792** | **0.454** |
>
> The results reveal that incorporating LALoss into the visual decoding loss DecLoss substantially enhances decoding performance.
>
> **Q5: Why is MindBridge only compared on a single entry? Additionally, why is there no visual comparison with MindBridge?**
>
> We provide detailed MindBridge results for each subject in Tab. 1, as well as a comparison of the reconstruction results between our method and several other approaches, including MindBridge, in Fig. 1 in the anonymous link (https://mindaligner.github.io/MindAligner/).
>
> **Reference**
>
> 1. Haxby et al. A common, high-dimensional model of representation in human visual cortex. PNAS 2011.
> 2. Naselaris et al. Encoding and decoding in fMRI. NeuroImage 2011.

---

> > ### Comment · Reviewer_5B7Z · 2025-04-06
> >
> > Thanks for addressing my concerns.
> >
> > Although the proposed solution, a linear mapping, appears simple at first glance, as noted by other reviewers, the authors are in fact introducing a new paradigm for cross-subject brain decoding: explicit alignment. In the rebuttal, the authors provide both neuroscience motivations and strong experimental evidence supporting the superiority of the linear hypothesis over other alternatives. This justifies their final choice of a linear-based solution. So in my view, this contribution is meaningful enough to warrant acceptance at ICML.
> >
> > Based on the above, I have decided to raise my score to Accept.
> >
> > That said, I strongly encourage the authors to include key rebuttal clarifications in the revised paper, particularly the justification for the linear assumption and the implications of explicit alignment, to enhance clarity and impact. This will make the final version much clearer and more compelling to readers.

---

> > > ### Author Response · Authors · 2025-04-07
> > >
> > > Thank you very much for your kind and encouraging response, as well as for recognizing the contribution of our explicit alignment method. We sincerely appreciate your valuable suggestions and will incorporate the key clarifications from the rebuttal to ensure the revised version is clear and compelling.

---

### Decision · Program_Chairs · 2025-05-01

**Decision:**

Accept (poster)

**Comment:**

Reviewers generally regarded the paper favourably and found the approach to be novel, well-motivated and to provide useful neuroscientific insights. In particular, the papers strengths include:
* The cross-subject brain decoding approach with explict functional alignment is well motivated and well-founded, and addresses limitations in current cross-subject methods
* The quantitative results show substantial advantages to this method across a broad set of metrics over the baseline in most cases
* The need for paired stimuli is relaxed with the soft alignment mechanism, which is novel and promising
* The interpretable functional alignment analysis facilitates insights into cross-subject variability and underlying neural mechanisms.

Limitations and concerns for the paper include:
* Some descriptions could be clearer, including the choice of some notation, the description of the method, conditions for experiments, the meaning of some elements of the results, and procedures for selecting samples for qualitative comparisons.
* A better discussion of the ablation results with regards to the latent alignment loss, and whether this is less critical when other auxiliary losses are included, would strengthen this aspect of the paper.
* Some indication of the significance or error bounds for quantitative results would strengthen the paper

The authors are strongly encouraged to include key rebuttal clarifications in the revised paper, including the justification and discussion of the linear assumption and the implications of explicit alignment.